# Exploring the Relationship between Formal and Informal Institutions, Social Capital, and Entrepreneurial Activity in Developing and Developed Countries

**Diana Escandon-Barbosa** [1,*] **, David Urbano-Pulido** [2] **and Andrea Hurtado-Ayala** [3]

[1]   Business Department, Pontificia Universidad Javeriana, Cali 16000, Colombia
[2]   Economía i empresa Department, Autónoma of Barcelona University, 08193 Barcelona, Spain;
      urbanopdavid@gmail.com
[3]   Business Studies Faculty, Antonio Jose Camacho, Cali 16000, Colombia; anday490@yahoo.com
*   Correspondence: dmescandon@javerianacali.edu.co; Tel.: +57-2-321-82-00

**Abstract:** Most research on entrepreneurial activities and institutions focuses on identifying certain relationships between formal and informal institutions and entrepreneurship across economies. In this study, we advance entrepreneurship research by examining how social capital as a characteristic of the institutional environment affects the relationship between formal and informal institutions and entrepreneurial activities, differentially, in developing and developed economies. Supporting institutional theory and social capital theory, the results from our sample of 39 countries from 2001 to 2014, which contains over 30,000 identified individuals, indicate that social capital has a stronger influence in the relations between institutions and entrepreneurship. In developing countries, this influence is greater in the relationship between property rights, access to credit, subjective insecurity, and entrepreneurial activity. In developed countries, the greater effect of social capital is on the relationship between corruption and entrepreneurial activity.

**Keywords:** formal institutions; informal institutions; social capital; entrepreneurial activity in developing and developed countries

## 1. Introduction

Several studies agree that the institutional context influences entrepreneurial activity [1–3], and in developing economies, these institutional conditions differ and lead to different aspirations at the enterprise level compared with in developed economies [2]. Prior research has posited the effects of different types of institutions on entrepreneurial activity, analyzing in particular the differences between formal and informal institutions [4–8], but few studies have analyzed the influence of different institutional contexts. Therefore, there is a gap in the study of how different institutional scenarios affect business growth and how this contributes according to the type of economy. Specifically, there is a need to understand how characteristics of the institutional environment of a country, such as trust, networking, and cooperative norms, may or may not favor the action of formal and informal institutions. In that regard, social capital may provide unique insights into the important process by which formal and informal institutions affect entrepreneurial activities. The analysis of social capital refers to the structures of social organizations such as social networks and common norms of benefits [9,10]. Social capital is built by means of personal resources accumulated through time, allowing the establishment of relational ties [11]. At the same time, such resources can be used for individual progress or to accomplish social objectives [12].

Social capital has also been crucial in understanding entrepreneurial processes [13], because it addresses the weaknesses of institutions, which often prevent them from ensuring the right conditions for starting a business [2,14]. Afandi, Kermani, and Mammadov [15] pointed out that despite studies that claim to have made significant progress in the analysis of social capital and its relation to entrepreneurship, there are certain limitations in the empirical research. First, previous studies have focused on the context of a single country and thus do not provide results that can be generalized to a wider geographic context [16,17]. Second, the measurement of capital is usually limited, as it is based on certain indicators that do not contemplate all its dimensions, revealing results that would not be describing the real role of social capital on entrepreneurship.

This paper also aims to respond to these gaps in two ways. First, the analysis is performed for a broader geographic context by including a comparison between developed and developing countries. This arises because, in developing economies, the institutional conditions differ from the entrepreneurial aspirations, just as they are different with respect to developed economies [18,19]. Second, to measure the concept of social capital, we employ the World Values Survey (WVS) database, which contains eight indicators that are used to measure the three dimensions of social capital [20]—norms, trust, and networks.

In this sense, this article develops a conceptual model in which we analyze formal institutions with regards to property rights and access to credit, and informal institutions with regards to corruption and subjective insecurity. We do this by comparing developing and developed countries and including the moderating role of social capital, taking into account that the latter differs to a great extent between countries [21] and, therefore, can operate differently in various institutional contexts.

The current research makes contributions to the literature by suggesting that the use of social capital plays an important role for business growth, as entrepreneurs make use of it to respond to institutional deficiencies [19]. In addition, it allows us to recognize which types of institutions are most affected by social capital, whether formal or informal, and helps us to understand the differences according to the level of development of the country. The hypotheses are tested using a logistic regression model on panel data. A hierarchical model is used for developed and developing countries in the 2001–2014 period, using the databases of the World Bank, the Index of Economic Freedom, the Global Entrepreneurship Monitor (GEM), and the World Values Survey (WVS).

The paper is structured as follows. Section 2 discusses the theory and develops the hypotheses. Section 3 describes the data and measurement of variables. Section 4 estimates the lineal hierarchical model, and Section 5 presents the results. Section 6 develops the discussion, and Section 7 Robustness Checks and Section 8 draws conclusions.

## 2. Theory and Hypotheses Development

### 2.1. Multidimensional Concept of Social Capital

Social capital theory explains how interpersonal networks allow access to social sustenance and how they affect the strengthening of norms and social behavior. This theory confirms that knowledge appears from the collaboration of people [22] in order to build larger networks with profound personal implications.

The origin of the term arises with Bourdieu [23], who defined it as the total of real or potential resources that are related to the existence of a network of lasting, recognized, and institutionalized relationships. This concept is part of a classification of three dimensions of capital—economic, cultural, and social. According to Bourdieu [23], social capital is a resource that appears in different areas or social fields. Each of these social fields differs by importance of three dimensions of capital. Social capital has two components; namely the connection of actors with social networks and the symbolic capital, or objective differences, between groups that make possible the recognition and distinction [23,24]

Subsequently, authors such as Coleman [25], Nahapiet and Ghoshal [12] and Tsai and Ghoshal [26] described social capital as an asset linked to relationships among individuals, organizations, and societies. Coleman [25] identified three dimensions of social capital—trust, networks and information channels, and civic norms and effective sanctions. By contrast, Nahapiet and Ghoshal [12] and Tsai and Ghoshal [26], according to this perspective, also pointed out that social capital is a three-dimensional construct—structural, cognitive, and relational. Structural social capital refers to a general configuration between organizational actors and the means by which these are interconnected to access a set of knowledge [27] Cognitive social capital suggests that the interrelations among actors is created within the social structure, and it increasingly strengthens the existing social capital. Finally, relational social capital is associated with the main normative dimensions, allowing mutual trust among different actors [9].

Similarly, Putnam et al. [9] defined social capital as "those characteristics of a social organization, such as trust, norms, and networks, which can improve the efficiency of society by facilitating coordinated actions". Although trust lacks a single concept, Paxton [28] suggested two types of trust—trust in people and in institutions. These two types of trust help reduce uncertainty and enable transactions to be carried out, promoting cooperation among individuals, organizations, and institutions [29]. Norms are habits that contribute to differentiating between acceptable and unacceptable behavior [15,30]. By contrast, civic norms are rules of conduct that are not written, but are reflected when participating in social activities where public values and interests are emphasized [31]. These types of rules favor consensus among people and seek to improve the welfare of society. Networks arise as a civic commitment that occurs through meetings with friends, family, professionals, associations, and so on. [32]. By means of the networks, communication channels are formed to obtain information, which increases cooperation. Given these definitions, this study assumes this multidimensional perspective of social capital [20].

## 2.2. Entrepreneurial Activity, Institutional Context, and Social Capital

Bourdieu (1986) and Putnam et al. [9] contribute to understanding entrepreneurship as a social practice. According to Mckeever, Anderson, and Jack [33]. from the perspective of these authors, it is recognized that social capital has an important influence on entrepreneurship. The social capital characterized by social networks provides the context within which entrepreneurs are embedded, allowing entrepreneurs to cooperate with each other, favoring business growth.

According to this, the context of entrepreneurs involves different contacts and resources that change in each phase of the entrepreneurial process, because during this process, people initially may not know what kind of personnel and institutions can help them, but in more advanced stages, the entrepreneur may act more selectively with connections compared with nascent entrepreneurs [34]. Moreover, high trust in individuals reduces the uncertainties in contracts [35] This relationship between entrepreneurial activity and social capital helps us to understand the influence of institutional trust in entrepreneurship. The effects of institutions on entrepreneurial activity may influence institutional trust because they can allow or prevent individuals from enterprising [15]

Social capital and its main dimensions have been analyzed in different disciplines (such as economics, sociology, law, and education) [11,33,36] and at the micro and macro levels. The relation between social capital and institutions is evident at the macro level, where the strength of social networks is mainly the result of the political, legal, and institutional environment [37]. The ability of social groups to act together depends on the stability of formal institutions that exist in the entrepreneurial ecosystem or country context [10]. For example, in emerging economies, there is low social capital because economic transactions are greatly influenced by private interests or individual relationships [38]. Cleaver [39] affirmed that institutional incapacity increases extra costs for all economic activities and creates negative effects from the collaborative activities that come from society.

This level has been analyzed in recent articles, and their contribution confirms that social capital does have an effect on growth [40]. However, it is necessary to analyze what determining

factors drive growth [41]; among these are high-quality institutions and entrepreneurship education. According to Kim and Kang [42], one of the less studied mobilizing factors of economic growth is entrepreneurship. Kwon and Arenius [43] suggested that social capital at the regional/country level drives entrepreneurial activities in that area, allowing to demonstrate that individual activities are affected by their close relationships with network ties and a broader social environment.

According to Koellinger [44], although developing countries have high imitating entrepreneurial activity, they still can manage to obtain some sort of economic benefit. Additionally, Koellinger [44] supported the view that developing countries have a high effect on the accessibility of opportunities to create new businesses to the extent that some countries have a different distribution of factors of production within society, allowing entrepreneurial activity to take place without innovation. Bosma and Levie [45] pointed out that entrepreneurial activity and its perception of opportunity is higher in efficiency-driven economies than in innovation-driven economies. Alvarez et al. [46] found evidence for Latin American countries (factor-driven and efficiency-driven economies) and concluded that the socioeconomic context (unemployment, lower education level, etc.) has an influence on entrepreneurial activity.

Other studies have confirmed that developed countries allow access to resources and the development of networks [47]. Social networks are the most invaluable resource that an entrepreneur has; they facilitate the detection of when and where to start an entrepreneurial activity, as well as how to find the available resources (human, financial, etc.) [14].

However, previous entrepreneurial experiences have influenced the development of social capital to the extent that networks help to decrease the difficulties that are present from the start [14]; they increase the availability of resources [48] as well as the opportunities for discovery [49]. For instance, Nahapiet and Ghoshal [12] argued that social capital and networks generate positive conditions to discover entrepreneurial opportunities through the creation of new knowledge. Networks help to access information and build knowledge in order to approach an entrepreneurial opportunity [50].

## 2.3. Formal Institutions and Entrepreneurial Activity: The Moderating Role of Social Capital

According to Verheul et al. [51], entrepreneurship has been strengthened by the institutional approach, arguing that the social and cultural contexts determine the final decision to start a business. Institutions are the rules of the game that regulate the economic, social, and political relationships in a society [52]. There is a distinction between formal and informal institutions [52]. The formal institutions are regulations; contracts; and political, legal, and economic rules that limit individual behavior. Gnyawali and Fogel [53] identified four institutional dimensions from the point of view of entrepreneurship: (1) government policies and procedures, with institutions such as property rights, business freedom, and labor freedom; (2) socioeconomic conditions, including the attitudes of society and governments toward entrepreneurial activities; (3) entrepreneurial and business skills, such as education, training programs, and so on; and (4) financial and nonfinancial assistance, emphasizing access to credit, financial freedom, and other aspects. In this paper, the formal institutions analyzed are property rights and access to credit.

One of the key elements of corporate activity is property rights, which constitutes an important formal institution that offers the appropriate incentives to entrepreneurs. Property rights avoid expropriations against the entrepreneur [54]; this encourages the creation of companies because the entrepreneur has greater protection for his operations [55].

It is also recognized that entrepreneurial activity tends to be affected by financial limitations. In most cases of enterprise creation, entrepreneurs tend to need financing to get their business started; for this reason, access to financing tends to involve one of the formal institutions with more influence over entrepreneurial activity [5].

2.3.1. Property Rights

In general, an appropriate protection of property rights for all types of entrepreneurship contributes to the growth of commercial activities, as it decreases the investment risk [56] and promotes innovatory behavior [57].

For Acemoglu and Johnson [58], two important aspects must be taken into account regarding property rights; that is, expropriation risk, whether it is due to the arbitrariness of the government or other actors, and the quality of institutions in charge of guaranteeing the protection of property rights. In this sense, the limitations of the government in regard to its capacity to confiscate wealth ensure the protection of property rights and thus favor entrepreneurship [59]. Thus, the restriction to the arbitrary use of power by politicians guarantees the protection of property rights, encouraging entrepreneurship [60].

However, social capital contributes to the protection of property rights that matter for entrepreneurial activity [61]. The trust supports important determinants of the market expansion activities by solving collective action problems, providing an environment associated with secure property rights, and improving economic performance [61,62]. According to the definition of Putnam [9], social capital comprises the features of a social organization, such as trust, norms, and networks, that improve the efficiency of society; thus, social capital increases the economic efficiency of property rights because it contributes to overcoming the limitations of a formal institution, such as property rights, that are rules configured to control economic interaction [63]. This relation between property rights and social capital encourages long-term contracting and is essential for the creation of firms and economic development [63]. In other words, social capital affects entrepreneurial activity via the property rights channel, but this relationship is more feasible in developing countries where formal institutions are less efficient [64]. For Ahmad and Hall [61], it is reasonable to expect that social capital would have possibly caused economic growth in developing countries to a greater extent than property rights, because although in these countries, the formal institutions such as property rights are lacking, the economies continue to grow. Therefore, the following hypothesis is proposed.

**Hypothesis 1 (H1).** *Social capital moderates the relationship between the protection of property rights and entrepreneurial activity, having a greater effect in developing countries than in developed countries.*

2.3.2. Access to Credit

There are limitations on access to credit depending on the type of activity to be developed; the amount of necessary capital to start activities; and the activity's risk level, including the location of the company; among others [65,66]. Therefore, greater access to credit positively affects the creation of new companies and the expansion of existing ones [55]. Nevertheless, entrepreneurs tend to start their business activities with their own capital, given the limited credit that financial institutions tend to offer to this type of entrepreneurship [67]. Cardoza et al. [68] suggested that there is limited availability of financial resources to encourage corporate activity in Colombia. In general, favorable conditions to grant credit are given to big companies, while banks are not willing to finance new and small companies, considering that these do not have guarantees and possess liquidity restrictions [69]. In the face of this situation, small and new companies must resort to financial sources by means of their families or personal contacts, or they must access banking credit through the bribing of employees of the banking institutions [70]. Shoji et al. [71] pointed out that poor social capital causes low trust in partners, and this possibly leads to less access to informal credit because the entrepreneurs suffer from low trust toward making business with others. This negative relation may suggest the possibility of social capital moderating the relationship between access to credit and entrepreneurship.

One of the options to access credit in emerging economies is by means of aspects such as trust in social networks that make available informal credit sources [72,73], because social capital improves credit market accessibility through social collateral mechanisms [74]. Hence, entrepreneurs in emerging countries can access other sources of credit through social capital, which favors the creation of firms. On this basis, the following hypothesis is proposed.

**Hypothesis 2 (H2).** *Social capital moderates the relationship between access to credit and entrepreneurial activity, having a greater effect in developing countries than in developed countries.*

*2.4. Informal Institutions and Entrepreneurial Activity: The Moderating Role of Social Capital*

According to North [52,75], informal institutions refer to traditions, values, beliefs, social norms, and practices that define a society's culture. Compared to formal institutions, informal institutions operate at a more tacit level, where both elements work in conjunction; while formal institutions regulate economic activities, informal institutions shape the perceptions and judgments of the self, others, and their environment [76]. The literature on entrepreneurship has emphasized the importance of informal relationships in society, because these shape the propensities of social groups toward entrepreneurship [48].

The World Bank measures informal institutions, including various factors of governance such as the process of electing governments, the capacity of the government to formulate and implement policies, and the perception of citizens about the state of institutions [77].

For their part, Douhan and Henrekson [78] proposed that the inefficiency of institutions may affect entrepreneurship, including corruption as an informal factor that can affect the entrepreneur's perception and his or her motivation to create a new business. In some emerging economies, violence is an informal institution because it causes a significant decline in the performance of firms [79]. Insecurity is an additional factor affecting entrepreneurship, and its negative effects increase the costs of investing in new businesses [80].

In this paper, we analyze corruption and subjective insecurity as informal institutions. Corruption is defined as the inappropriate use of public power to obtain private benefits [81,82]. Corruption presents a negative impact on a country's economy to the extent that it impedes its development, slows down the creation and enacting of social policies that favor citizens, hinders public funds, increases the salary gap, and creates a negative image that disincentivizes foreign investment [81,83,84]. By contrast, subjective insecurity is the perception that each individual builds about the risk level faced in the city [85].

2.4.1. Corruption

Meanwhile, corruption decreases the motivation of the entrepreneur to create a new business because of the high transactional costs that must be incurred [86]. In addition to this, corruption can favor consolidated companies that might take advantage of public resources to increase their benefits, discouraging the formation of new businesses [81].

Corruption also tends to appear in the institutional social context [87]. In countries where people tend to move away from legal norms, any illegal behavior becomes normal as long as it is accepted and followed by many people [88]. In this way, corrupt behavior may become a social norm, and an individual might carry out unlawful business if there is a high business-based perception that other businessmen do it as well [28]. However, if the control of corruption considered as an informal institution is analyzed, this tends to improve the entrepreneur's incentive to create new businesses [89]. Now, a greater control of corruption tends to be present in developed countries [90] while in a developing country, the control tends to be deficient, and thus corruption tends to have a pronounced negative effect on the entrepreneurial activity [91].

Analyzing each of the components of social capital, according to Mitchell [92] networks are constituted by information, exchange, and expectations; thus, social networks are frameworks for evaluating the behavior of people in different situations [93]. High levels of trust in institutions and compliance to norms can lower practices of corruption [87,94]. Finally, if anticorruption values are strongly shared within society, this will be more likely to result in an incentive for entrepreneurial activity [59,87]. Additionally, the networks act as a monitoring agent of public administration. This behavior introduces better managerial practices that affect organizational growth [87].

Social capital could be a good explanatory factor for the reduction of corruption in developing and developed countries—a situation that would improve the probability of starting new firms. However, the social context and the policies for eradicating corruption usually tend to be more feasible and more common in developed countries [87]. Aidt [95] contended that in developing countries, efficient corruption arises to facilitate the trade between agents that is not possible otherwise. In these countries, the corruption is good for economic growth because the institutions are poor, and corruption reduces bureaucracy [96]. However, the control of corruption is lower in developing countries than in developed countries, and the impact of social capital also tends to be smaller. On this basis, the following hypothesis is proposed.

**Hypothesis 3 (H3).** *Social capital moderates the relationship between corruption control and entrepreneurial activity, having a greater effect in developed countries than in developing countries.*

2.4.2. Subjective Insecurity

A favorable social environment is an incentive for entrepreneurship. Problems with violence and lack of safety are barriers to entrepreneurship, and trust and loyalty among individuals are fundamental [80]. The perception of citizen insecurity requires more spending on the part of entrepreneurs, who must invest in safeguarding their businesses from the negative effects of violence and insecurity. The perception of insecurity is common in developing countries. Economic and social inequality are the main causes of insecurity and conflict. In this context, it can be difficult to do business, particularly formal business activities [3].

However, one of the ways to improve the perception of insecurity is through social capital. Inter-group interactions can help to resolve conflict and increase the welfare of countries [97]. Developing countries are constrained in the ability to invest in all forms of capital; thus, increases in social capital are stimulated to combat the threat of insecurity [97]. Several forms of social capital, such as social networks and trust, can help individuals to protect themselves from situations of insecurity. According to Sawyer [98], neighborhood watch schemes began in the United States in the 1970s and were exported to other countries. Such schemes were also created in developing countries as a community response against thefts and other crimes; therefore, this relationship is usually higher in developing economies, where the perception of insecurity is greater than in developed economies. Finally, the development of social capital to combat insecurity leads to individuals making use of a greater number of networks and taking advantage of them to increase entrepreneurial activity based on the discovery of new opportunities [99]. Therefore, the following hypothesis is proposed.

**Hypothesis 4 (H4).** *Social capital moderates the relationship between subjective insecurity and entrepreneurial activity, having a greater effect in developing countries than in developed countries.*

## 3. Materials and Methods

*Sample and Data Collection*

We tested four hypotheses about the relationship between formal and informal institutions and entrepreneurial activity in developing and developed countries. This paper uses an unbalanced panel of 39 countries (Annex 1) for the years 2001 through 2014.

The sample uses data available in the GEM survey, which contains over 30,000 identified individuals (other studies with GEM [100,101]). Additionally, these data are complemented by those of the WVS and the World Bank in order to obtain information about social capital and the variables of control. These measurements will be explained in detail in the following sections.

## 4. Measures

Dependent variable: Our dependent variable is total entrepreneurial activity (TEA) taken from the GEM survey. Nowadays, the GEM data set is highly recognized as an important source of information for analysis of entrepreneurship because it collects data from different countries and builds comparative results [89,102,103]. Through a common methodology for all countries, entrepreneurs and their environment are described. This contrasts with other databases, which include limited countries for which the measurements differ. Since 1999, GEM has collected information for numerous aspects of entrepreneurship and thus shows indicators of entrepreneurial activity.

TEA measures the level of entrepreneurial activity in each country from 2001 to 2014. Moreover, to meet our research objectives, two groups of countries were created, developed and developing, taking into account the United Nations Human Development Index (HDI), which serves as a reference to identify those countries with high levels of development. The TEA is calculated as the percentage of the adult population between 18 and 64 years of age who have initiated a new business or own or lead a young business.

Independent variables: The social capital variable is measured on the WVS survey. Eight indicators are used to measure the three dimensions of social capital [9] norms, trust, and networks. The data were grouped into tracts of five years each. For the period analyzed in this paper, we looked at the 1999–2004, 2005–2009, and 2010–2014 surveys. The WVS is carried out through face-to-face surveys in 43 countries and collects political, cultural, economic, and behavioral information. The first step was to perform a factorial analysis on the WVS. This analysis included the trust, norms, and network variables contained in the survey. This method of data reduction is justified so as to accomplish a more robust measure of the social capital concept [104] The total of the sample was 83,017. The factorial analysis used the method of main components, and all items were reduced to a single factor that accomplished a charge of over 0.60 [105].

All variables were not available for all years and countries. According to Tabellini [106] this situation is not problematic given that cultural values shift slowly through time. In this sense, social capital had low variation on almost all of the analyzed countries in the WVS survey over the studied term.

By contrast, the variables of property rights, corruption, and subjective security are measured by the World Bank, and we used the available data between 2001 and 2014. Lastly, data for access to credit are from the GEM database, with data available for all the studied years.

Control variables: For this study, the ages of the population and economic growth variables were included from the World Bank database. These variables have been used in previous research to control the TEA [89]. We considered the total population between the ages of 15 and 64, representing the number of people who could potentially be economically active. Economic growth rates are measured as gross domestic product (GDP) growth at purchase prices, represented by the sum of gross value added by all resident producers in the economy plus any product taxes and minus any subsidies not included in the value of the products.

Table 1 summarizes the variables, definitions, and sources.

**Table 1.** Description of the variables.

| | Variable | | Definition | Source [a] |
|---|---|---|---|---|
| **Dependent Variable** | Total entrepreneurial activity (TEA) | | Total early-stage entrepreneurial activity. Percentage of adults aged 18–64 setting up a business or owning/managing a young firm (up to 3.5 years old), including self-employment. | GEM 2001–2014 |
| **Independent Variables** | Social capital (moderator variable) | | The proxies of social capital are as follows: trust in others, participation in groups or associations, the index of civic cooperation, confidence in public services, confidence in political institutions, confidence in armed forces and police, and confidence in empowering institutions [107]. | WVS 1999–2004; 2005–2009; 2010–2014 |
| | Formal institutions | Property rights | Property rights protection across judicial systems against theft and expropriation. | IEF 2001–2014 |
| | | Access to credit | Sufficient debt funding available in the country for new and growing firms. | GEM APS 2001–2014 |
| | Informal institutions | Control of corruption | Reflects perceptions of the extent to which public power is exercised for private gain, including both petty and grand forms of corruption, as well as "capture" of the state by elites and private interests. | WGI 2001–2014 |
| | | Subjective insecurity | Perception of insecurity, perceptions of the likelihood of political instability and/or politically motivated violence, including terrorism. | WGI 2001–2014 |
| **Control Variables** | Population age (PA) 15–64 | | Total population between the ages of 15 to 64, representing the number of people who could potentially be economically active. | WGI 2001–2014 |
| | Economic growth rate (EGR) | | Gross domestic product (GDP) growth at purchase prices is the sum of gross value added by all resident producers in the economy plus any product taxes and minus any subsidies not included in the value of the products. | WGI for the period 2001–2014 |

[a.] Sources World Governance Indicators (WGI) by World Bank, http://databank.worldbank.org/data/home.aspx; Global Entrepreneurship Monitor (GEM), http://www.gemconsortium.org; index of economic freedom (IEF), http://www.heritage.org/index; WVS (World Values Survey), http://www.worldvaluessurvey.org/wvs.jsp.

## 5. Estimation Method

The data were analyzed with a lineal hierarchical model between 2001 and 2014, with the dependent variable TEA and independent variables—property rights, access to credit, subjective insecurity, and social capital. This model was performed on developed and developing countries.

Lineal hierarchical models allow the inclusion of effects associated with two or more levels. Our case allows the combination of effects associated with the country and individuals. According to Liu and Gupta [108], these models also allow the performance of progressive adjustments by steps in order to know the relevance of each independent variable in explaining the dependent variable.

In this paper, we propose a model with three levels [109,110].

Model 1: Model with variables of control. This level aimed to test the variance that existed between countries on the dependent variable for the period between 2001 and 2014.

Model 2: Model with control and principal variables. In this phase, we analyzed the annualized effects of each variable—social capital, property rights, access to credit, corruption, and subjective insecurity.

Model 3: Model with control, main, and moderation variables. This last model permits the inclusion of all the variables under analysis, including the main effects of each variable and the interaction variables. These variables are included because of the acknowledgement that social capital affects TEA, but also boosts its effects and is in contact with other institutional factors, both formal and informal.

In mathematical terms, the final model would be as follows [110,111]:

$$Y_{ij} = \beta_{0j} + \sum_{n=1}^{3} \beta_{ij}\chi_{nij} + \beta_{3}\chi_{3j} + \sum_{n=4}^{m} \beta_{1}\chi_{nj} + \sum_{n=m+1}^{l} \beta_{n}\chi_{nj} + \varepsilon_{ij}$$

Level 1: $\beta_{0j} = \beta_0 + \mu_{oj}$; $\beta_{1j} = \beta_1 + \mu_{1j}$; $\beta_{2j} = \beta_2 + \mu_{2j}$; $\beta_{3j} = \beta_3 + \mu_{3j}$
Level 2: $Z = \chi \times y$; $Z2 = \chi2 \times y2$; $Z3 = \chi3 \times y3$

The mode was calculated by aleatory effects, where it is assumed that the countries were taken from a larger population, thus allowing the generalization of the effects to all groups of countries. In this sense, the aleatory effects permit the acceptance of the existent variation among countries in regard to the degree of entrepreneurial activity (TEA). Hence, a mean of 0 was assumed, with a variance that is constant and correlated with the rest of the covariables [112].

## 6. Results

Table 2 reports the mean, standard deviations, and correlation of the variables used in this study for developed and developing countries. TEA is correlated with all the formal and informal institutions of this study. Once the correlations were obtained, we tested the multicollinearity for each group through variance inflation factor (VIF). The VIF values are low in all models—control, main effects, and the model with moderations. The highest VIF is 1.52 in the model with moderations, but this is still far from the acceptable limits.

**Table 2.** Descriptive statistics and correlation matrix.

| Variables | Mean | Std. Dev. | Mean | Std. Dev. | 1 | 2 | 3 | 4 | 5 | 6 | 7 | 8 |
|---|---|---|---|---|---|---|---|---|---|---|---|---|
| TEA | 3.589 | 4.1 | 2.612 | 0.345 | 1 | | | | | | | |
| Property rights (PR) | 3.195 | 17.42 | 4.511 | 8.228 | 0.1262 | 1 | | | | | | |
| Access to credit (AC) | 4.135 | 0.975 | 3.167 | 0.291 | 0.4454 | 0.0295 | 1 | | | | | |
| Corruption (C) | 1.221 | 0.964 | 4.416 | 0.123 | −0.3879 | −0.1896 | 0.1233 | 1 | | | | |
| Subjective insecurity (SI) | 3.23 | 0.562 | 5.636 | 1.26 | −0.49 | 0.125 | 0.1456 | 0.0145 | 1 | | | |
| Social capital (SC) | 2.4534 | 1.964 | 1.242. | 1.123 | −0.1317 | 0.0082 | 0.2321 | −0.1134 | −0.1082 | 1 | | |
| Population age 15 to 64 | 3.23 | 0.562 | 5.636 | 1.26 | −0.3341 | 0.2295 | −0.4935 | −0.1483 | 0.5163 | −0.648 | 1 | |
| Economic growth rate (EGR) | 3.23 | 0.562 | 5.636 | 1.26 | −0.1241 | 0.01495 | −0.5235 | −0.2683 | 0.5163 | −0.648 | 0.012 | 1 |

Table 3 shows the general results of the hierarchical models applied for developed and developing countries. The first model was applied with the control of the variables, then the main effects were included, and finally the effects of the moderation were included. In Table 3, we can see that for both groups, the complete model (Model 3 for both types of countries) has better adjustment indicators related to the change in F that is significant ($p < 0.00$), allowing us to conclude that it is the best model for both groups.

**Table 3.** Adjustment statistics.

| | Model | R | R Squared | R Squared Adjusted | Standard Error of Estimation | Change in R Squared | Change in F | Sig. Change in F |
|---|---|---|---|---|---|---|---|---|
| Developing country | 1 | 0.589 [a] | 0.346 | 0.342 | 0.969 | 0.346 | 77.125 | 0 |
| | 2 | 0.635 [b] | 0.403 | 0.392 | 0.931 | 0.056 | 9.039 | 0 |
| | 3 | 0.657 [c] | 0.432 | 0.416 | 0.913 | 0.03 | 4.938 | 0.002 |
| Developed country | 1 | 0.619 [a] | 0.362 | 0.358 | 0.929 | 0.362 | 79.252 | 0 |
| | 2 | 0.682 [b] | 0.433 | 0.430 | 0.942 | 0.070 | 10.055 | 0 |
| | 3 | 0.711 [c] | 0.451 | 0.448 | 0.943 | 0.18 | 6.927 | 0.000 |

[a] Predictor: (Constant), PA, EGR; [b] Predictor: (Constant), PA, EGR, PR, AC, C, SI, SC; [c] Predictor: (Constant), PA, EGR, PR, AC, C, SI, SC, PR × SC, AC × SC, C × SC, SI × SC; Dependent variable: TEA.

Table 4 shows the results of the aleatory-effect lineal regression model. Model 1 includes only control variables, and the TEA is considered as a function of population age (PA) and economic growth rate (EGR). By contrast, Model 2 estimated the control and principal effect. The TEA is a function of

PA and EGR and the main effects of formal and informal institutions, such as property rights, access to credit, corruption, subjective insecurity, and social capital.

Finally, in Model 3, both of the previous blocks of variables were included, with the addition of social capital moderating formal and informal institutions. All the models are significant and a high explanatory power was obtained, explaining more than 68% of the TEA variance for both types of countries.

The third model for developing and developed countries confirms that all informal institutions have statistical significance and affect entrepreneurial activity ($p < 0.005$).

Concerning the hypotheses testing, Hypothesis 1 proposed that social capital positively moderates the relationship between the protection of property rights and entrepreneurial activity, having a greater effect in developing countries than in developed countries. The coefficient for developed and developing countries is positive and significant (developed countries: b = 0.15620, $p < 0.01$; developing countries: b = 0.17461, $p < 0.01$).

Therefore, not only is entrepreneurial activity influenced by property rights, but it also positively influences social capital. The results also show that the coefficient in developing countries is higher in comparison with developed countries, supporting Hypothesis 1 (t = 3.15, $p > 0.01$). This may be explained by the fact that in developing countries, the construction of social capital serves to improve the absences or flaws in the regulation of property rights, becoming a powerful strategy to improve TEA.

Hypothesis 2 suggests that social capital positively moderates the relationship between access to credit and entrepreneurial activity, having a greater effect in developing countries than in developed countries. This hypothesis is supported by our data; the presence of opportunities to access credit in conjunction with social capital affects entrepreneurial activity (developed countries: b = 0.1636, $p < 0.01$; developing countries: b = 0.1861, $p < 0.01$), and the results also show that the coefficient in developing countries is higher in comparison with developed ones (t = 2.16, $p > 0.01$), in line with the literature [72,73].

**Table 4.** Estimating TEA with formal and informal institutions.

| (Constant) | Control Model | | Principal Effect Model | | Model with Moderations | |
|---|---|---|---|---|---|---|
| | Developed Countries | Developing Countries | Developed Countries | Developing Countries | Developed Countries | Developing Countries |
| | Coefficient | Coefficient | Coefficient | Coefficient | Coefficient | Coefficient |
| Property rights (PR) | | | 0.1429 ** (4.834) | 0.1321** (3.941) | 0.2256 *** (17.12) | 0.247 *** (19.351) |
| Access to credit (AC) | | | −0.258 ** (−1.158) | −0.221 ** (−1.021) | −0.2095 *** (18.560) | 0.192 *** (16.32) |
| Corruption (C) | | | −0.1656 ** (−2.037) | −0.1541 * (−1.967) | −0.0981 (4.112) | −0.1112 *** (5.86) |
| Subjective insecurity (SI) | | | 0.01539 * (1.986) | 0.01345 * (1.896) | −0.1531 *** (6.896) | 0.1871 *** (8.501) |
| Social capital (SC) | | | | | 0.1428 *** (6.745) | 0.1856 *** (8.458) |
| PR*SC | | | | | 0.15620 *** (5.756) | 0.17461 *** (7.8947) |
| AC*SC | | | | | 0.1636 *** (4.5453) | 0.1861 *** (5.253) |
| C*SC | | | | | −0.0754 *** (2.19) | −0.10018 *** (2.45) |
| SI*SC | | | | | 0.1314 *** (3.756) | 0.1499 *** (4.87) |
| Population age (PA) | 0.12270 ** (2.81) | 0.1345 ** (2.93) | 0.0201 (1.18) | 0.0001 (1.10) | 0.0012 (0.125) | −0.0315 (0.673) |
| Economic growth rate (EGR) | −0.16257 *** (12.377) | 0.234 *** (4.345) | −0.0024 * (−1.650) | −0.0019 (−1.450) | −0.0028 (1.590) | −0.0443 (0.820) |

Significance: * 0.10; ** 0.05; *** 0.01.

Hypothesis 3 suggests that social capital positively moderates the relationship between corruption control and entrepreneurial activity, having a greater effect in developed countries than in developing countries. This hypothesis is also confirmed (developed countries: b = −0.0754, $p < 0.01$; developing

countries: b = −0.10018, $p < 0.01$) (t = 3.12, $p > 0.01$); the existence of social capital helps to control the negative effects of corruption on entrepreneurial activity.

Hypothesis 4 suggests that social capital positively moderates the relationship between subjective insecurity and entrepreneurial activity, having a greater effect in developing countries than in developed countries. This hypothesis is confirmed for developed and developing countries (developed countries: b = 0.1314, $p < 0.01$; developing countries: b = 0.1899, $p < 0.01$) (t = 3.29, $p > 0.01$). As with the previous hypothesis, the relevant role of social capital is confirmed as a social construction of mechanisms to increase the levels of entrepreneurial activity (TEA).

Graphics 1 and 2 show the moderating effect of social capital for the relationship between TEA and each of the dependent variables included in the model (property rights, access to credit, corruption, and subjective insecurity) for both developed and developing countries. The y-axis of each graph represents the values obtained for TEA for different values of social capital in developing countries (Figure 1) and for developed countries (Figure 2) introduced in the estimated regression function. On the x-axis, the different values of each dependent variable are presented.

In Figure 1, we can see that the values indicated inside the graph represent the effect of property rights (PR), access to credit (AC), corruption (C), and subjective insecurity (SI) on TEA for each social capital (SC) level. Furthermore, to compare the significance of the moderating effect of SC, which is the object of the hypotheses, it is verified in all cases because the change in the effect between low and high levels is significant. (PR × SC: $\beta$alto−$\beta$bajo = 5.22; $t = 4.86$; $p < 0.01$; AC × SC: $\beta$alto−$\beta$bajo = 4.562; $t = 4.62$; $p < 0.01$; C × SC: $\beta$alto−$\beta$bajo = 2.002; $t = 1.92$; $p < 0.01$; SI × SC $\beta$alto−$\beta$bajo = 1.802; $t = 1.99$; $p < 0.05$).

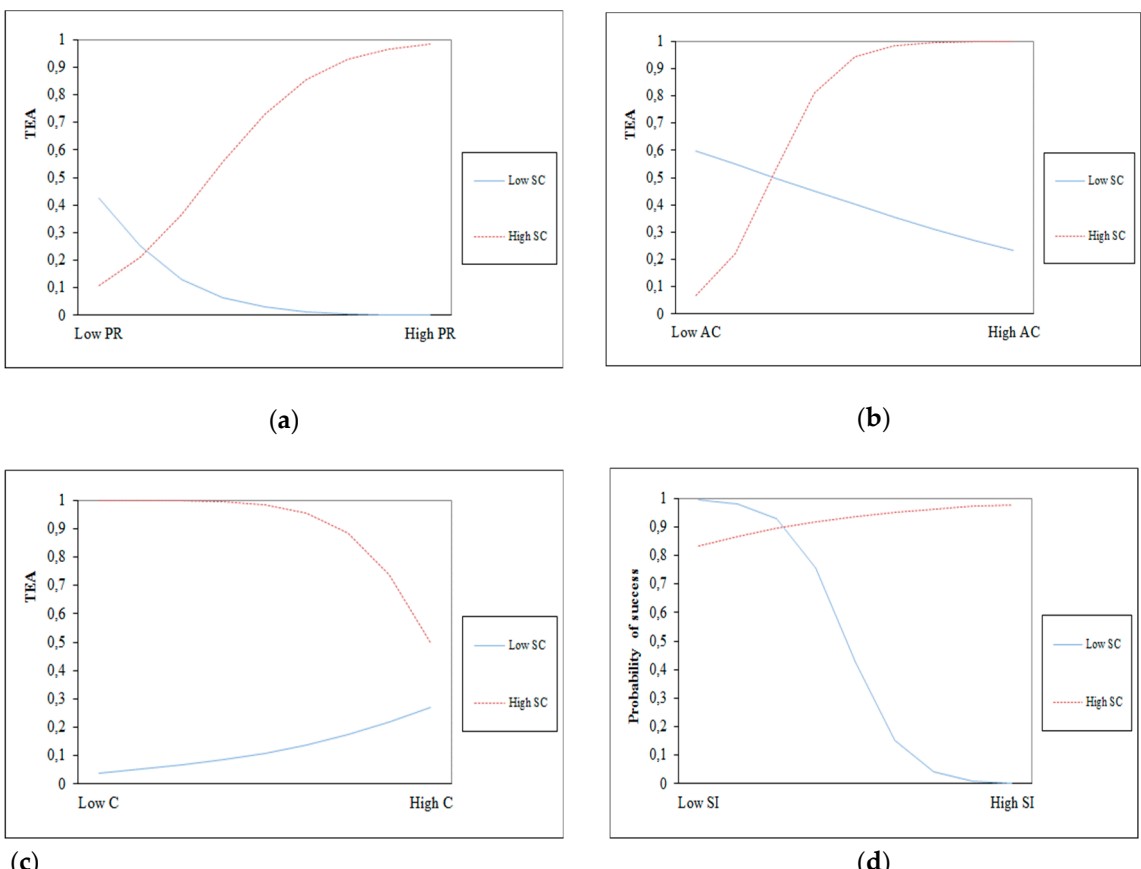

**Figure 1.** Moderation Effects developed countries. TEA—total entrepreneurial activity; (**a**) PR—property rights, SC—social capital; (**b**) AC—access to credit, SC—social capital; (**c**) C—corruption, SC—social capital; (**d**) SI—subjective insecurity, SC—social capital.

In the case of Figure 2, it can be observed that the significance for the four hypotheses is also fulfilled (PR × SC: $\beta$alto−$\beta$bajo = 5.42; $t$ = 4.91; $p$ < 0.01; AC × SC: $\beta$alto−$\beta$bajo = 6.034; $t$ = 5.41; $p$ < 0.01; C × SC: $\beta$alto−$\beta$bajo = 1.002; $t$ = 1.98; $p$ < 0.01; SI × SC $\beta$alto−$\beta$bajo = 1.502; $t$ = 2.09; $p$ < 0.01).

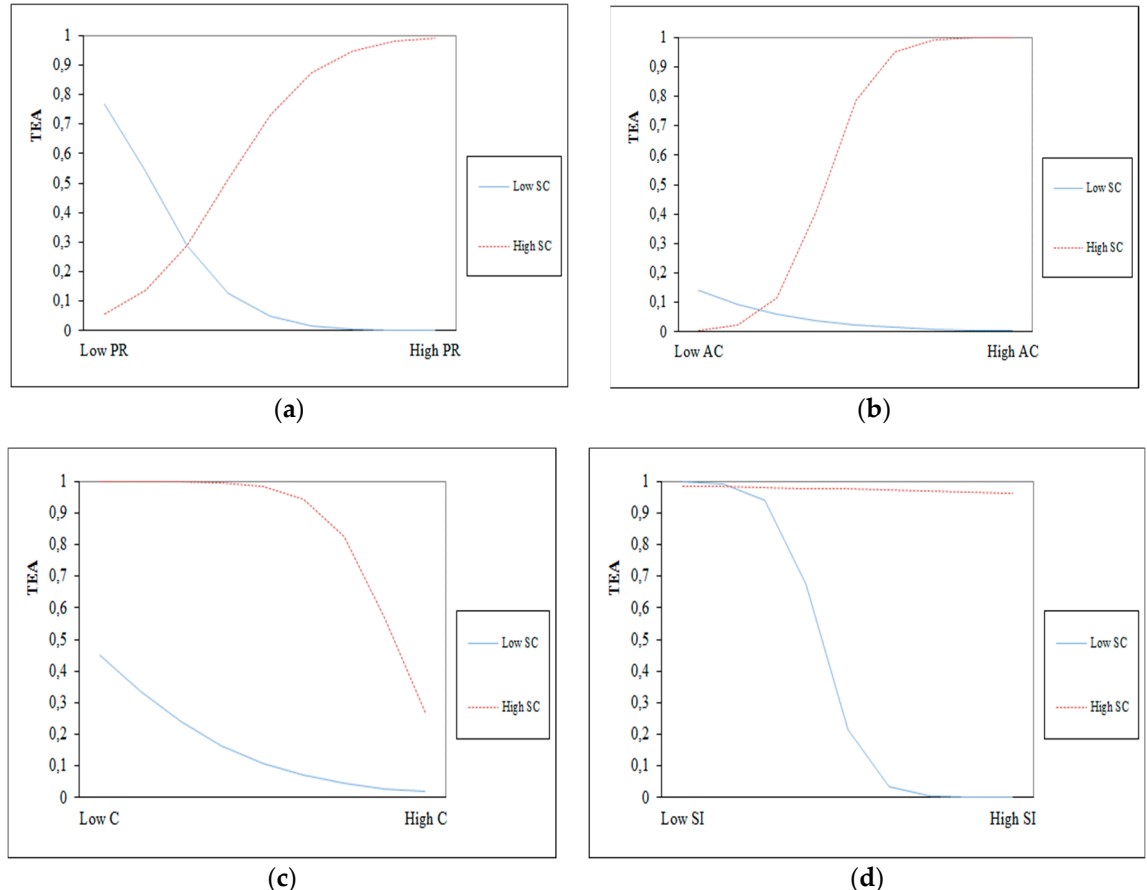

**Figure 2.** Moderation Effects developing countries. TEA—total entrepreneurial activity; (**a**) PR—property rights, SC—social capital; (**b**) AC—access to credit, SC—social capital; (**c**) C—corruption, SC—social capital; (**d**) SI—subjective insecurity; SC—social capital.

Through graph analyses, this study allowed us to verify that high levels of social capital produce better levels of entrepreneurial activity (TEA); in other words, social capital is a highly effective strategy to counteract the negative effects of corruption and subjective insecurity, and it enhances the effect of access to credit for entrepreneurship and protection of property rights.

These positive effects are seen in developing countries, where the social conditions provided by the construction of social capital achieve good results in TEA in conjunction with existing formal and informal institutions.

In the case of developed countries, the effect of mediation is significant for all cases, but only when it moderates the relationship between corruption and TEA can it have a greater effect in these countries compared with developing countries, as suggested by Hypothesis 3. For the other institutions, the opposite effect is achieved. These results suggest the possibility of improving the rate of entrepreneurial activity based on the creation of social trust, civic norms, and behavior that can be highly accepted and successful in developing countries where the absence of formal institutions and control entities is pronounced.

## 7. Robustness Checks

We performed robustness checks on our results. Historical time series analyses can have different problems or reach limited conclusions because of the change in associated time structures. Our analysis covered a 14-year time period. We utilized a jackknife resampling technique to check the robustness in our results. This technique is convenient for estimating the robustness of these types of models [113]. The results confirm that sub-survey estimations (jackknife) are very comparable to the original results—providing assurance for the original estimations.

## 8. Conclusions

The current study offers opportunities to enrich the entrepreneurship literature. Specifically, we extend our understanding of social capital based on the approaches of institutional theory and social capital to argue for the moderating role of social capital on the relationship between formal and informal institutions and entrepreneurial activity, comparing developed and developing countries. The findings allow us to comprehend three important aspects. First, social capital is understood as an informal institution given that it encompasses certain aspects of a country's or region's social conditions that are perceived by its inhabitants, such as trust in people and institutions, trust in social networks, and civic norms or rules of behavior. Second, social capital intervenes in the relationship between formal and informal institutions and entrepreneurial activity, being regarded as characteristic of the social organizations that contribute to better functioning of such institutions. Third, differences regarding social and economic conditions between developed and developing countries cause social capital's input to differ, tending to be positive in those countries where the weakness of formal and informal institutions is stronger, which means that in developing countries, social capital works as another kind of informal institution that permits the combating of the weakness so as to favor entrepreneurial activity.

The results support the evidence found in other papers referenced in this article. Social capital improves a society's efficiency and, therefore, the rules that configure economic activity. Among them are those that protect property rights, achieving a greater impact in developing countries, where fewer controls and guarantees for rights exist [64].

In developed countries, access to credit is more favorable for entrepreneurs; these countries have economic stability and thus sound financial leverage, whereas in developing countries, given the process of development they face, there are weak financial institutions and policies to finance the creation of new businesses. Given the above, social capital plays an important role in these economies given that personal or family credit sources tend to prevail over access to financial institutions. Trust and the creation of social networks tend to be the route of access to informal credit, which allows the development of entrepreneurial activity [72]. A similar situation occurs with informal institutions. Developing countries usually have less control of corruption, and there is lower trust in institutions and their efficiency. In this case, social capital tends to be more relevant in developed countries, where trust in institutions is higher. Thus, social capital usually improves the context of corruption in the promotion of entrepreneurial activity, but its impact is greater when dealing with economies where there are strong anti-corruption policies, as in the case of developed countries [87] Something similar happens with the perception of insecurity, which tends to be greater in developing countries. Many of these countries face problems of violence and criminality, which go against the interests of creating new companies. For this reason, social capital may help to counteract such a perception of insecurity to a higher degree in developing countries, where, despite the perception of insecurity by their inhabitants, trust among people and other norms of legitimate behavior may counteract subjective insecurity and motivate entrepreneurial initiatives [97].

From these results, it may be concluded that social capital allows the exploration of other channels by means of which a country's institutional weaknesses may be counteracted, seeking to favor economic development by means of entrepreneurship. Because of this, the main implication is centered on the opportunity that countries have to take advantage of their citizens' perception of societal functioning

so as to improve the general performance of institutions. It is worth highlighting the importance that informal institutions have in those countries where formal institutions are especially weak, as people are able to rely on informal institutions to regain trust and boost the creation of new businesses that contribute to economic development. Finally, the findings of this research imply that trust, relationships with others, and norms of civic cooperation as components of social capital are more effective in developing countries than in developed countries, because the latter have stronger institutions as well as greater control of corruption. In developing countries, social capital would reinforce such control.

Our research offers practical implications as well. The findings reported in this study show that one of the main functions of policy makers is the promotion of policies to entrepreneurs. This function must be fulfilled in all countries. Our results show that if policy makers want to stimulate entrepreneurial activity, they must discover ways to increase social capital through plans, strategies, rules, and norms. Although social capital is a collective construction or informal institution, policy makers can help in developing a social integrated environment inside cities and countries. Additionally, promoting social capital may be good for policy makers in the long run. High social capital always brings benefits to TEA through property rights and credit access. At the same time, it reduces the negative impact of corruption and subjective insecurity. This is highly important especially for developing countries, where formal institutions are very weak and informal institutions like social capital can help improve entrepreneurial activity.

There are many possibilities for new studies based on these results, including studies where every component of social capital and individual contributions to the functioning of institutions and entrepreneurial activity may be specifically analyzed through empirical analysis at the country and regional level.

We surveyed 39 countries, but other countries face a different institutional environment, which may affect the impact of social capital on the relationship between formal and informal institutions with regard to entrepreneurial activities, as well as differences in developing and developed countries. Future research is needed to improve the country database and to examine the validity of our model in other countries and groups (e.g., for continents and regions).

In addition, entrepreneurial activities measured by our study can be improved by taking entrepreneurial behavior into account. We propose including the notion of levels of entrepreneurship rather than classifying individuals as entrepreneurs or non-entrepreneurs. In doing so, we can examine differences over time across countries.

Data availability is a common limitation, given that it is not easy to access a single database for most countries with the necessary variables and time spans. It was for this reason that a panel of data was not estimated, because while the WVS database is governed by tracts of years, other databases measure the variables annually. While this limitation was overcome in this study, it would be interesting for future research to resolve this through the construction of surveys by country, which would contain these types of variables, at least initially, at the country level.

**Author Contributions:** D.E.-B.: methodology and results; D.U.-P.: ideas, revisions, and literature; A.H.-A.: literature.

**Funding:** This research received no external funding only received internal funding (Javeriana University).

**Acknowledgments:** The authors are grateful to the GEM Colombia for database and Jairo Salas for Ubg and Tap.

**Conflicts of Interest:** This research does not have any conflicts of interest.

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
