# Peer review of "Exploring the Relationship between Formal and Informal Institutions, Social Capital, and Entrepreneurial Activity in Developing and Developed Countries"

_sustainability, doi:10.3390/su11020550_

Round 1

Reviewer 1 Report

Very well written and interesting paper.   Solid empirical work.  If possible, I would not mind seeing a reference or two to prior scholarship in this journal.  It would be nice to place this piece in context to other work done in this journal.   And then hopefully later articles in this journal will cite your research! 

Overall, I was quite impressed.

page 5 “property rights” section.  It would be worth the authors’ time
to take a look at and cite Hernando De Soto - why capitalism triumphs in
the west and fails everywhere else.

some of the table formatting seemed inconsistent - usage of bold.  Table
1 “Property rights” “Access to credit” “Control of corruption” and
“Subjective security” probably does not need to be bold.

page 9 lines 414-415.  The boxes with arrows did not line up.
  Presumably the production process can take care of this, but please
double check an be be careful - otherwise it is not logical.

Table 2 - Headings in the column under “Variables” probably do not need
to be bold.

page 15 “we surveyed only 39 countries , but other countries…” good
point.  maybe delete “only”  otherwise this is a decent statement.

references - I’m not sure, so see style guideline, but I would have
thought reference 7  would have first letters of book title capitalized
  The Art and Science of Entrepreneurship instead of The art and science
of entrepreneurship.

reference 14 looks like there should be a space AsiaPacific Journal of
Management
reference 15 don’t think you need “the” before American Political
Science Review

reference 24 - capital first letter of words in book title - I think,
but maybe not.
reference 26    semicolon after insurance before using.

reference 33 evolotionARY
reference 40 theory _&_ practice  - be consistent

reference 47 delete “the” before QJE
reference 49 “and” replace with & or whatever you decide - be consistent
- even if the journal is not!

a few other similar issues throughout references which you may catch or
the copy editors will.

Author Response

We appreciate the comment of the reviewer and share with him/her the weakness aspect mentioned

some of the table formatting seemed inconsistent - usage of bold.  Table
1 “Property rights” “Access to credit” “Control of corruption” and
“Subjective security” probably does not need to be bold.

This words was modify

page 9 lines 414-415.  The boxes with arrows did not line up.
  Presumably the production process can take care of this, but please
double check an be be careful - otherwise it is not logical.

This boxes was modify

Table 2 - Headings in the column under “Variables” probably do not need
to be bold.

Tks. This headings was modify

page 15 “we surveyed only 39 countries , but other countries…” good
point.  maybe delete “only”  otherwise this is a decent statement.

This Word was modify

references - I’m not sure, so see style guideline, but I would have
thought reference 7  would have first letters of book title capitalized
  The Art and Science of Entrepreneurship instead of The art and science
of entrepreneurship.

reference 14 looks like there should be a space AsiaPacific Journal of
Management
reference 15 don’t think you need “the” before American Political
Science Review

reference 24 - capital first letter of words in book title - I think,
but maybe not.
reference 26    semicolon after insurance before using.

reference 33 evolotionARY
reference 40 theory _&_ practice  - be consistent

reference 47 delete “the” before QJE
reference 49 “and” replace with & or whatever you decide - be consistent
- even if the journal is not!

Tks. This references was modify

Reviewer 2 Report

This is a well written paper which will be of interest to the journal audience. The paper draws upon the relevant literature for a paper of this type. The study identifies gaps in the literature to achieve a novel contribution to knowledge in this field. Appropriate hypothesis are generated from the extant literature. The statistical model looks fit for purpose with an appropriate research design. The sample size is excellent. However, the methodology section would benefit from greater justification from the academic literature as to where these variables and data have been previously used in prior studies. The results section look appropriate and fit for purpose. The conclusions section is well developed with consideration of contribution achieved, implications for key stakeholders, study limitations and further research opportunities. The final reference list should be consistently presented with all journal papers appearing with both volume and issue numbers present.

In the literature section the following issues need to be addressed:

On page 3 the sentence beginning "Structural social capital...." requires a supporting academic reference.

On page 3, avoid using the expression "On the other hand" use "By contrast"  instead. This happens on more than one occassion. See also page 6.

On page 3, "Finally, relational social capital....." also requies a supporting academic reference.

On page 3, "By means of the networks..." the following sentence requires a supporting reference.

On page 3 do not start a sentence with the word "Also" use "In addition" or "Moreover" as an alternative.

On page 3, the sentence beginning "The ability of social groups " requires a suppoting reference.

On page 6, the sentence "Corruption also tends to appear..."  requires a supporting reference.

The use of GEM data for analysis is acceptable however some precedent in its usage should be provided. So provide references to where GEM data has been used in other studies. For example see Small Business Economics or recent Journal of Business Research studies.

Similarly, the use of the variables within the analysis model should be further justified with reference to prior literature. For example, TEA has been extensively used in previous research studies see JoBR.

In the final reference list any journal should appear with both Volume and Issue number identified. Currently this is inconsistent.

Author Response

In the literature section the following issues need to be addressed:

On page 3 the sentence beginning "Structural social capital...." requires a supporting academic reference.

On page 3, avoid using the expression "On the other hand" use "By contrast"  instead. This happens on more than one occassion. See also page 6.

On page 3, "Finally, relational social capital....." also requies a supporting academic reference.

On page 3, "By means of the networks..." the following sentence requires a supporting reference.

On page 3 do not start a sentence with the word "Also" use "In addition" or "Moreover" as an alternative.

On page 3, the sentence beginning "The ability of social groups " requires a suppoting reference.

On page 6, the sentence "Corruption also tends to appear..."  requires a supporting reference.

This sentences was modify

The use of GEM data for analysis is acceptable however some precedent in its usage should be provided. So provide references to where GEM data has been used in other studies. For example see Small Business Economics or recent Journal of Business Research studies.

We has been added  a country´s table.

Similarly, the use of the variables within the analysis model should be further justified with reference to prior literature. For example, TEA has been extensively used in previous research studies see JoBR.

Tks. We add reference to prior literature

In the final reference list any journal should appear with both Volume and Issue number identified. Currently this is inconsistent.

Tks. This mistake has been  identified

Reviewer 3 Report

I have two questions to Authors:

Your period of research is 2001-2014. Is it 14-year or 13-year time period? According to you it is 13-year (line 515).

Your sample contains 39 countries divided into two parts: developing and developed. Could you please add a small table presenting names of countries in each part?

Best regards

Author Response

Your period of research is 2001-2014. Is it 14-year or 13-year time period? According to you it is 13-year (line 515).

Tks, We has been changed this mistake

Your sample contains 39 countries divided into two parts: developing and developed. Could you please add a small table presenting names of countries in each part?

Yes. We has been added this table